# Intrauterine Growth Restriction—Prediction and Peripartum Data on Hospital Care

**DOI:** 10.3390/medicina59040773

**Published:** 2023-04-16

**Authors:** Marina Dinu, Andreea Florentina Stancioi-Cismaru, Mihaela Gheonea, Elinor Dumitru Luciu, Raluca Maria Aron, Razvan Cosmin Pana, Cristian Marius Marinas, Stefan Degeratu, Maria Sorop-Florea, Andreea Carp-Veliscu, Andreea Denisa Hodorog, Stefania Tudorache

**Affiliations:** 18th Department, Faculty of Medicine, University of Medicine and Pharmacy of Craiova, 200349 Craiova, Romania; marina.dinu@umfcv.ro (M.D.);; 2Obstetrics and Gynecology Department, Emergency County Hospital, 200349 Craiova, Romania; 31st Department, Faculty of Medicine, University of Medicine and Pharmacy of Craiova, 200349 Craiova, Romania; 4Obstetrics and Gynecology Department, Targu-Jiu County Hospital, 210218 Targu-Jiu, Romania; 5Department of Obstetrics and Gynecology, Carol Davila University of Medicine and Pharmacy, 020021 Bucharest, Romania; 6Panait Sarbu Clinical Hospital of Obstetrics and Gynecology, 060251 Bucharest, Romania; 7Obstetrics and Gynecology Department, Mioveni City Hospital, 115400 Mioveni, Romania

**Keywords:** early-onset intrauterine growth restriction, late-onset intrauterine growth restriction, ultrasound, estimated fetal weight, tertiary care center, prenatal care, hospitalization days

## Abstract

*Background and Objectives:* We aimed to prospectively obtain data on pregnancies complicated with intrauterine growth restriction (IUGR) in the Prenatal Diagnosis Unit of the Emergency County Hospital of Craiova. We collected the demographic data of mothers, the prenatal ultrasound (US) features, the intrapartum data, and the immediate postnatal data of newborns. We aimed to assess the detection rates of IUGR fetuses (the performance of the US in estimating the actual neonatal birth weight), to describe the prenatal care pattern in our unit, and to establish predictors for the number of total hospitalization days needed postnatally. *Materials and Methods:* Data were collected from cases diagnosed with IUGR undergoing prenatal care in our hospital. We compared the percentile of estimated fetal weight (EFW) using the Hadlock 4 technique with the percentile of weight at birth. We retrospectively performed a regression analysis to correlate the variables predicting the number of hospitalization days. *Results:* Data on 111 women were processed during the period of 1 September 2019–1 September 2022. We confirmed the significant differences in US features between early- (Eo) and late-onset (Lo) IUGR cases. The detection rates were higher if the EFW was lower, and Eo-IUGR was associated with a higher number of US scans. We obtained a mathematical formula for estimating the total number of hospitalization days needed postnatally. *Conclusion:* Early- and late-onset IUGR have different US features prenatally and different postnatal outcomes. If the US EFW percentile is lower, a prenatal diagnosis is more likely to be made, and a closer follow-up is offered in our hospital. The total number of hospitalization days may be predicted using intrapartum and immediate postnatal data in both groups, having the potential to optimize the final financial costs and to organize the neonatal department efficiently.

## 1. Introduction

Intrauterine growth restriction (IUGR) is a complex problem in modern obstetrics. The definitions of IUGR varied largely in recent decades, and a consensus was eventually proposed [1]. This particular group of fetuses, not reaching their biological growth potential due to impaired placental function, need specialized healthcare prenatally, intrapartum, and postnatally. Many require iatrogenic preterm delivery for fetal or maternal indications [2]. By consensus, the cut-off chosen for dividing the two different clinical scenarios (early-onset versus late-onset IUGR) is 32 weeks [3] of gestational age (GA). It seems that late-onset IUGR (Lo-IUGR) is more difficult to diagnose, but it has a lower association with perinatal hypoxic events and suboptimal long-term neurodevelopment [4]. Screening techniques applied for Lo-IUGR (to detect suboptimal fetal growth or reduced growth velocity) have the potential to diagnose restricted fetuses, but their differentiation from small-for-gestational-age fetuses (healthy and smaller than the majority) is challenging in many cases.

Although cost-effective, complete neonatal care for these fragile newborns is expensive and represents a burden for tertiary units. The direct costs of care for low-birth-weight infants, resulting from newborn intensive care unit (NICU) days, postnatal hospitalization, and the first year of life, are high [5]. The number of hospitalization days is an important predictor of the total costs of admitted cases.

Many studies were designed to establish the diagnostic markers (biometry and/or functional markers as Doppler studies) for IUGR and focused on prediction models for adverse postnatal outcomes. We used a modified definition of consensus in IUGR [1], as it is the best substitute for diagnosis in the prenatal period. Most probably, postnatal diagnosis (neonatal growth restriction) is as challenging as prenatal diagnosis. Defining neonates at risk based solely on size is unlikely to be sufficient. The relevant outcomes that should be reported in future trials are still debated. The relationship between IUGR and neonatal features needs to be evaluated further, and this study is intended to support this endeavor, providing current medical practice data from our hospital.

In this study, we aimed to report our prospectively acquired data on cases of IUGR: the demographic data of the mothers, the prenatal ultrasound (US) features, and the immediate postpartum data in a group of fetuses with IUGR cared for in our unit. 

The reported accuracy of growth US is moderate in detecting a low birth weight [6,7,8]. We aimed to assess the performance of prenatal US scans in detecting smallness in neonates, a function of the severity of growth abnormalities, in our setting. We compared the US estimated fetal weight (EFW) percentile using Hadlock nomograms with the birth weight percentile.

We aimed to describe the local prenatal care pattern by correlating the EFW percentile with the number of US scans scheduled in the third trimester (TT). 

We also established predictors for the number of total hospitalization days needed postnatally in our unit and provided a mathematical formula based on specific intra- and post-partum parameters.

## 2. Materials and Methods

This is a nested cohort prospective study, designed and conducted in the Prenatal Diagnosis Unit of the Emergency County Hospital of Craiova, a tertiary referral university-affiliated hospital in the southwest region of Romania. The study period was 1 September 2019–1 September 2022. The methodology of enrolment and partial results (the Eo-IUGR cohort) were previously published [9].

The data in this report encompass a larger cohort of Eo-IUGR cases since, in the mentioned study, we excluded all pregnancies with Eo-IUGR continuing at and later than 32 weeks’ gestation (WG). In this study, we describe the whole population of restricted babies, regardless of the timing of the diagnosis and the GA at delivery.

We enrolled women with suspected IUGR consecutively, using a lower threshold than the current consensus [1]: for Eo-IUGR (<32 WG), we used three isolated parameters (abdominal circumference (AC) < 10th centile, EFW < 10th centile, and absent end-diastolic flow in the umbilical artery—UmbA) and four parameters combined (AC or EFW <10th centile combined with a pulsatility index (PI) > 95th centile in either the UmbA or uterine artery—UtA). For Lo-IUGR (≥32 weeks), we used two isolated parameters (AC or EFW < 10th centile) and four parameters combined (EFW or AC < 10th centile, AC or EFW crossing centiles by > two quartiles on the Hadlock growth chart, and the cerebroplacental ratio (CPR) < 5th centile or UA-PI > 95th centile). The CPR was calculated by dividing the PI of the MCA by that of the UmbA [10] at every scan in TT.

The study included singleton pregnancies with accurately established GA, meeting the criteria at any scan between 22 and 38 WG, and having no known structural or genetic abnormalities. We used the Hadlock 4 technique for the US estimation of the fetal weight [11]. Patients with unavailable or incomplete data were excluded from the analysis.

We used a Voluson E10 (GE Medical Systems Chicago, IL, USA) US machine, equipped with a 4–8 MHz curvilinear transducer, in all cases. The study protocol was approved by the university ethics committee, and informed consent was obtained from all participants prior to enrolment. 

On the US scan form completed for this study, placenta (lateral) localization was added. The observer diagnosed “lateral placenta” if more than half of the placenta was seen on US on one side of the uterine cavity only. The corresponding (right or left) uterine artery was named “placental”. The other one was named the “non-placental” uterine artery on the US form. In cases with the placenta located rather centrally, the operator decided the assignment of the uterine arteries based on subjective criteria. The uterine spectral Doppler waves were interpreted objectively (by obtaining the pulsatility index—PI) and subjectively (protodiastolic notching was included in the US form).

The unit’s intern protocol was previously described [9]. The case management was customized based on the GA and on the results of US surveillance, including the Doppler findings in UtA, UmbA, and the middle cerebral artery (MCA); the fetal biophysical profile score; and the EFW [9]. The attending physician was able to increase or decrease the intensity of prenatal care and to decide the frequency of US scans. In internal policy, we offer a protective C-section in primigravidae with IUGR. Decisions on the timing of delivery were made by a dedicated multidisciplinary team, including at least two Ob-Gyn senior consultants and a neonatologist.

Counseling was extensive, and parents were involved in shared decision making. We have five complete neonatal intensive care units (NICUs), three local neonatal units (LNUs), and twelve special care baby units (SCBUs, low-dependency units). The hospital’s resources at a given time and the parental desire (e.g., transfer to another unit or fetal abandonment) were discussed and considered. 

Demographic data and maternal baseline characteristics, as well as data regarding pregnancy and newborn outcomes, were collected.

For interpreting the neonatal weight percentile, we used reference data from INTERGROWTH-21(st) Project [12] and the calculators offered at https://www.omnicalculator.com/health/birthweight-percentile [13], accessed between September 2019 and February 2023. 

We subsequently used statistical methods to find a formula for estimating the total hospitalization days needed postnatally based on the birth weight percentile, the number of NICU days, the Apgar score, and the GA at birth.

### Statistical Analysis

A statistical analysis was performed using Minitab 17 Statistical Software. As most quantitative data followed a non-normal distribution, medians and interquartile ranges are reported. For a comparison between groups, we used Mann–Whitney tests for non-normal data, and for categorical data/proportions, we used the Pearson Chi-squared test. We used a main effect plot to compare the detection rates for Eo-IUGR and Lo-IUGR based on percentile < 10 and the number of prenatal visits. To test the variance between groups, we used Levene’s test. The significance level was 0.05.

## 3. Results

One hundred eleven women were enrolled. According to the fetal US features and to the GA at enrolment, we collected data on 52 Eo-IUGR and 59 Lo-IUGR fetuses. 

The demographic and general maternal characteristics of the pregnant women in the study are presented and compared in Table 1.

The median maternal age was similar in the two groups, and the median BMI was lower in Lo-IUGR. In our cohort, the Eo-IUGR cases were more likely to be associated with GDM. None of the above-mentioned differences reached statistical significance. All clinical forms of hypertension related to pregnancy were more frequent in Eo-IUGR. In our cohort, smoking (including current and former smoking pregnant women) had a significant association with Eo-IUGR.

The ultrasound characteristics of the pregnancies included in the study are presented in Table 2.

The data presented in Table 2 were obtained at fixed GAs, 22–23 and 32–33 weeks’ gestation (WG). 

All US parameters examined showed significant differences in the two groups investigated. Both uterine arteries (placental and non-placental) assessed by means of spectral Doppler in the second trimester (ST) and in the TT reached significant differences, being higher in the Eo-IUGR group. The same was found when searching for the presence of uterine artery protodiastolic notching in the TT (the placental and the non-placental uterine artery). The CPR percentiles were lower in the TT in the Eo-IUGR group than in the Lo-IUGR group. The two investigated parameters of fetal biometry (the abdominal circumference and the EFW), expressed in percentiles, also differed statistically significantly in both trimesters. The mean gestational age at delivery was 30.7 (27–34) weeks in the Eo-IUGR group and 37 weeks (35–38) in the Lo-IUGR group. 

We noted and compared the mean placental weight, the GA at delivery, the total number of prenatal visits (all including US scans), and the number of visits in the third trimester in the two groups (Table 3). Placental weight is reported in grams, mean +/− standard deviations. 

All parameters showed statistically significant differences between the two groups.

In the Eo-IUGR group, we observed three cases of intrauterine fetal death (IUFD) (incidence of 8.1%); we registered no fetal demise (IUFD) in the Lo-IUGR group.

We confirmed that Eo-IUGR is associated with a higher number of prenatal visits (total and TT) when compared with Lo-IUGR. 

The newborn data in pregnancies complicated with Eo-IUGR and Lo-IUGR are presented in Table 4.

As expected, the postnatal NICU and total hospitalization days were significantly higher in the Eo-IUGR group, and the Apgar score was significantly lower. Resuscitation measures were required at birth in a higher percentage of neonates in the Eo-IUGR group and almost all neonates presented one or more episodes of transient apnea. In very few cases, hypotension occurred. Persistent ductus arteriosus (PDA) was diagnosed frequently in the Eo-IUGR group and rarely in the Lo-IUGR group. All cases of intraventricular hemorrhage were mild. Technical advances made the occurrence of hypothermia impossible. A single case of necrotizing enterocolitis occurred in the early group and received surgical treatment on the third day of life. Neonates in the early group had various degrees of anemia, and all received blood products or transfusions. This event was rarely seen in the Lo-IUGR group. All neonates developed jaundice, with most of them having minor forms. Transient hypoglycemia occurred in almost half of the Eo-IUGR cases immediately after birth and rarely in the late group. We observed no case of severe persistent hypoglycemia. Despite the routine empirical antibiotic treatment, we observed thirteen severe cases of neonatal infection. One only case had an early-onset form, while the remaining cases were diagnosed with late-onset infection.

We developed a main effect plot to see the interaction between prenatal detection, the EFW percentile, and the number of prenatal visits in the TT (see Figure 1). We demonstrated a higher prenatal detection when there was a lower EFW percentile. The prenatal detection rates reached 90–100% in cases with an EFW percentile ranging from 2 to 4%, and they were lower in cases with a higher EFW percentile. The figures in the latter situation varied from 80% to 30% in subjects with an EFW percentile between 5% and 10%. We also proved that the number of prenatal visits and US scans proposed and scheduled by the medical team in the TT was higher, against the EFW centile variation. The number of registered prenatal visits ranged between 5 and 15 in the TT in low-EFW cases and was lower in cases with a higher EFW percentile (ranging from 2 to 5). 

We performed a test for equal variance to compare the EFW percentile and birth percentile in both groups (see Figure 2). For the Eo-IUGR samples, we assumed an equal variation at *p* value > 0.05 (Levene’s test *p* = 0.240). So, in the Eo-IUGR group, the EFW percentile (1–6%) was similar to the birth percentile (1–6%). For the Lo-IUGR samples, we found a significant standard deviation difference (unequal variance), as the p value for Levene’s test was less than 0.05 (*p* = 0.01). Thus, in the Lo-IUGR group, the EFW percentile (1–9%) was different from the birth percentile (1–15%).

We performed a regression analysis to see which variables affect the total number of hospitalization days. In this further analysis, we included all neonates falling at and under p10 in birth weight. We searched for correlations between the number of hospitalization days and the birth weight percentile, the NICU days, the Apgar score, and the gestational age at delivery. We provide the final regression model as follows: Hospitalization days = 47.10–95.7 newborn percentile + 1.4127 NICU days + 2.883 Apgar Score − 1.571 GA at delivery. 

We found strong correlations using this formula (95% confidence, *p*-value less than 0.01, Pearson coefficient R-sq 83.18%). R-Sq showed that IUGR (defined as a birth weight less than the 10th percentile for the given gestational age), the number of NICU days, the Apgar score, and the gestational age explain 83% of the variation in the number of hospitalization days (see below for the statistical analysis—Figure 3).

The regression results show that birth weight under p10, the number of NICU days, the Apgar score, and the GA are significant because of their low p-values. The regression explains 83.18% of the variance of hospitalization days based on birth weight under p10; the number of NICU days; the Apgar score; and the GA, expressed in weeks. 

## 4. Discussion

Not much progress has been made in recent decades in the standardization of IUGR management, according to one of the most prominent researchers in this field, who agrees that the timing of the delivery of an IUGR fetus is still a critical issue: ”It is not possible through analysis of the available data, rationalization and/or argument to reach a conclusion about the relative merits of using venous Doppler, fetal heart rate analysis, the biophysical profile or a specific combination of these tests that will satisfy and persuade the proponents of each technique” (Romero et al., 2002, p. 2 [14]).

Moreover, there are discrepancies between the recommendations of the most important professional societies: Society for Maternal-Fetal Medicine (SMFM) [15], American College of Obstetricians and Gynecologists (ACOG) [16], and American Institute of Ultrasound in Medicine (AIUM) [17] suggest not to use a Doppler assessment of the DV, MCA, or UmbA for the routine clinical management of IUGR, whereas the International Society of Ultrasound in Obstetrics and Gynecology (ISUOG) guidelines recommend its use [18].

IUGR reflects deleterious fetal growth in an unfavorable environment due to an abnormal placenta. The best modality/best association of modalities to assess the fetus, to make the IUGR diagnosis, and/or to decide on delivery are still under research. Formally, if defective, the placenta seems to be involved in the occurrence of both clinical forms of IUGR. The placenta is responsible for both the nutrition and respiration of the developing fetus. Malplacentation affects both these functions. The fetal nutritional and respiratory needs are GA-dependent, yet conversely [19]. Respiratory demands seem to increase exponentially. In Eo-IUGR, fetal nutrition and growth are predominantly compromised. In Lo-IUGR, the respiratory function is more threatened. Thus, the natural history in Eo-IUGR is much longer, with fetuses surviving for weeks before requiring delivery. In contrast, Lo-IUGR, especially at term, affects fetal respiratory needs. Therefore, later in pregnancy, fetuses are more vulnerable, and the US diagnosis is more difficult. Sudden fetal demise in fact has an underlying (at times unrecognized) respiratory dysfunction. This may occur before the growth abnormality occurs. In IUGR, smallness is specific to the Eo form and is more rarely seen in the Lo scenario. 

Our data are intended to enlarge the area of knowledge, highlight the correlation between prenatal and postnatal features, and help in optimizing resource distribution in our hospital and similar units. It may be used in future logistic and linear regression analyses performed to identify independent predictors for the long-term outcomes of restricted fetuses. 

It is a common belief that many avoidable stillbirths are related to the failure to detect growth abnormalities antenatally. It has been reported that the detection of ”small for gestational age” (SGA) and IUGR remains suboptimal, ranging from 10% [20,21,22] to 54% [23]. We have recent evidence that routine US screening at 35–36 weeks’ gestation using the 10th percentile as the cut-off in the Hadlock 4 formula is moderately effective, predicting 63% of SGA neonates [24].

TT, AC, and EFW seem to perform similarly in predicting smallness for gestational age [25], with a small advantage in detection when using the AC. In our study, the AC and EFW differed significantly between the two groups (early- versus late-onset IUGR).

It was previously reported that severe smallness (an EFW under p3) has a much higher risk of adverse perinatal outcomes, irrespective of the Doppler pattern in Lo-IUGR [21]. Our study confirms that prenatal detection is better if the EFW percentile is lower; thus, US is more valuable within the most vulnerable population in the targeted group.

It seems that universal screening (maternal characteristics plus EFW) has low detection rates in IUGR (around 65%) at 30–34 weeks, with better figures in severe IUGR of less than one centile [21,26]. Sovio found that introducing TT routine US at a GA between 28 and 36 tripled the detection rate in nulliparous women. Yet, the group only reached a detection rate of 57%. There is evidence of better performance when IUGR is targeted. In our study, all fetuses falling under p10 in EFW were considered for enrolment. This is probably why we observed no unexpected small neonates and no IUFD in this cohort. The awareness of the team was increased in these cases.

There are studies reporting high discrepancies between EFW and birth weight, which might be difficult to explain due to the limitation of TT US, giving rise to the hypothesis of in utero fetal weight loss [27]. We proved that the Hadlock 4 formula performed better for the Eo-IUGR group, in which the EFW percentile was similar to the birth percentile. For the Lo-IUGR samples, we found a significant standard deviation difference (unequal variance). In this group, the performance of the formula was modest (the EFW percentile was different from the birth percentile).

It was previously reported that smoking in pregnancy is associated with IUGR, a low birth weight, and a higher UmbA PI after 34 weeks of gestation, but this association was not confirmed [28]. Our results are discordant. In our case series, smoking had a significant association with Eo-IUGR.

In our setting, prenatal care was more intense, conversely to the GA at diagnosis: the earlier the diagnosis, the higher the number of prenatal visits. In our cohort, the total number of prenatal visits in pregnancy and the total number of TT visits were higher than the numbers recommended by the current guidelines [18,29]. Although operator-dependent, the easily obtained parameter of protodiastolic notching may be useful in emergency settings and/or if using a suboptimal piece of US equipment with default settings. It was previously reported [30] that Dopplers do correlate with fetal size in severe IUGR and that Doppler changes might be related to absolute fetal AC size and not to growth velocity.

Our results are in line with the previously published link between early IUGR and hypertensive disorders in pregnancy. We confirmed this known association [3,31,32]. We found that hypertensive disorders were more likely to occur in Eo-IUGR cases than in Lo cases.

All fetal measurements and all Doppler values change with GA. All data should be converted into z-scores, percentiles, standard deviations, or multiple of medians (MoMs). We used percentiles for all Doppler parameters for convenience and homogeneity. In agreement with previously reported data [3,32], Lo-IUGR showed a less obvious alteration in UmbA and UtA Doppler flow.

In patients with Lo-IUGR, MCA Doppler may be useful in raising the awareness of health professionals but cannot be used for delivery timing; furthermore, in the surveillance of IUGR, it has a low predictive value for adverse perinatal outcomes [33]. CPR seems to have a poor predictive value for IUFD, but it seems better than the EFW in the prediction of adverse outcomes at birth [34]. We chose to use CPR centiles to describe and compare the IUGR fetuses.

The mean weight of the placenta was lower than the 10th percentile in the IUGR group, and the differences between the early- and the late-onset IUGR groups were statistically significant. These results are in line with those of previous reports [35,36].

As a limitation, in this study, the US expertise of the primary referring doctors was not investigated. In Romania, obstetrician ultrasonographers are the main healthcare provider responsible for the assessment of fetal growth in low-risk pregnancies. There is a high probability that the healthcare professionals’ skills and awareness also make a difference in detection rates and in customized management. In our country and in many other European countries, the TT US scans are optional, a function of the physicians’ and the woman’s preferences. It is accepted that time since medical authorization is often inversely associated with technical skills, knowledge, and adherence to new scientific-evidence-based protocols. Yet, experience is positively associated with diagnostic skills.

Moreover, diabetes, maternal prepregnancy weight (women with obesity, overweight, and underweight), and maternal weight gain were not assessed as confounders of EFW accuracy. The percent error is related to birth weight, and the clinical relevance of the over- and under-estimation of weight extremes is under investigation [37,38].

As advantages, using a single operator and a single piece of US equipment ensured the continuity of care, primarily related to the advantage of having the same pair of hands assessing fetal size and/or plotting the subsequent EFW on the same graphic curve. This facilitated the interpretation of growth velocity, which is very intuitive for doctors.

We intend to continue and scale up this study and to report the long-term follow-up data on children enrolled as patients.

In our view, our study results are not generalizable; however, they may have some practical implications. The higher the GA at birth, the shorter the NICU and total hospitalization stay and, consequently, the lower the costs (as well as the higher the survival rates). We provided a formula for estimating the total number of hospitalization days needed postnatally, based on birth weight, the number of NICU days, the Apgar score, and the GA. We found that these parameters explain 83.18% of the variance of hospitalization days. In our view, this information may help in resource distribution, and, moreover, it has the potential to improve the estimation of the number of neonatal units available at a certain time point.

Neonatal care is costly for these babies due to the technology involved and the prolonged lengths of stay. Our state hospital is suffering from fiscal pressures, like many others, especially in the post-COVID-19 era. To deal with such pressures, obstetricians, neonatologists, and hospital administrators must cooperate to wisely distribute the resources needed for inpatients postnatally: personnel costs, nursing costs, other support personnel, attending physician costs, hotel costs, equipment and ancillary costs (pharmacy, clinical laboratories, radiology, blood bank, respiratory therapy, electrocardiogram, electroencephalogram, occupational therapy/physical therapy, and disposable supplies (plastic catheters, syringes, tubes, gloves, etc.)). Our data may help improving targeted resources toward tertiary hospitals caring for this population of mothers and fetuses.

## 5. Conclusions

Early- and late-onset IUGR have different US features prenatally and different postnatal outcomes. If the US EFW percentile is lower, a prenatal diagnosis is more likely to be made. In our setting, a closer follow-up is more likely to be offered. The total number of hospitalization days needed postnatally may be predicted at birth, using intrapartum and immediate postnatal data, in both early- and late-onset IUGR. This has the potential to optimize the distribution of resources, planning, and the beforehand appraisal of the admission capacity in the neonatal department.

## Figures and Tables

**Figure 1 medicina-59-00773-f001:**
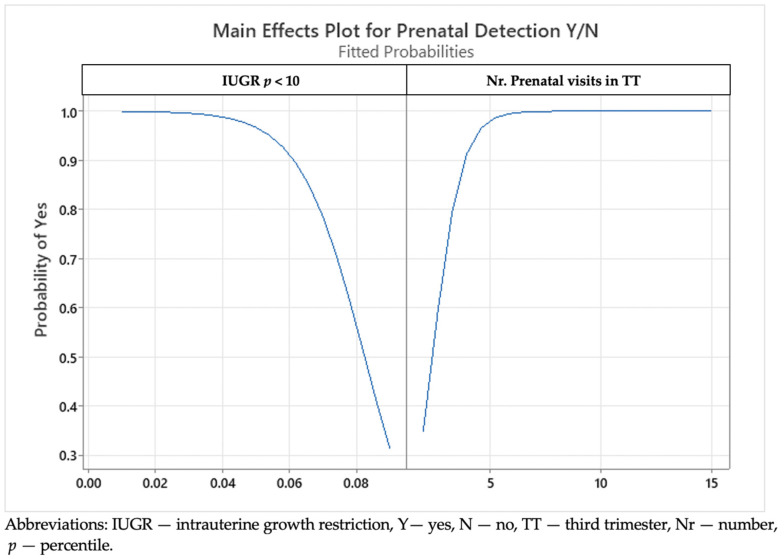
A main effect plot showing interactions between the prenatal detection, the estimated fetal weight percentile, and number of prenatal visits in the third trimester.

**Figure 2 medicina-59-00773-f002:**
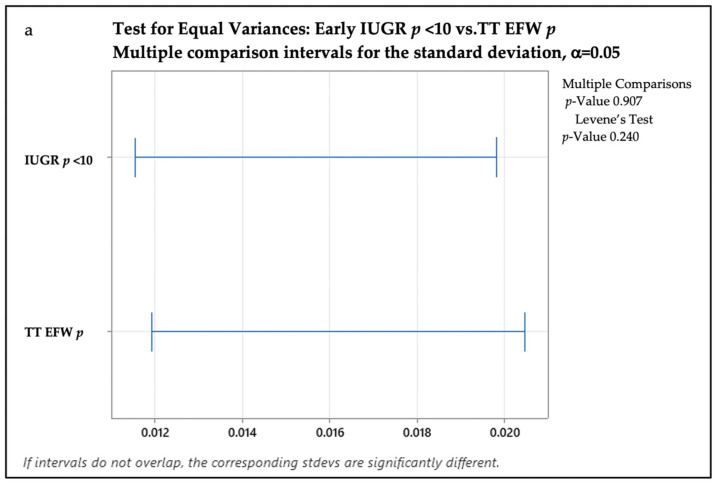
A graphic presentation of the test for equal variance to compare the estimated fetal weight percentile and birth percentile: (**a**) the test for equal variance to compare the estimated fetal weight percentile and birth percentile in the Eo-IUGR group; (**b**) the test for equal variance to compare the estimated fetal weight percentile and birth percentile in the Lo-IUGR group. Abbreviations: IUGR—intrauterine growth restriction; TT—third trimester; EFW—estimated fetal weight; *p*—percentile; stdevs—standard deviations.

**Figure 3 medicina-59-00773-f003:**
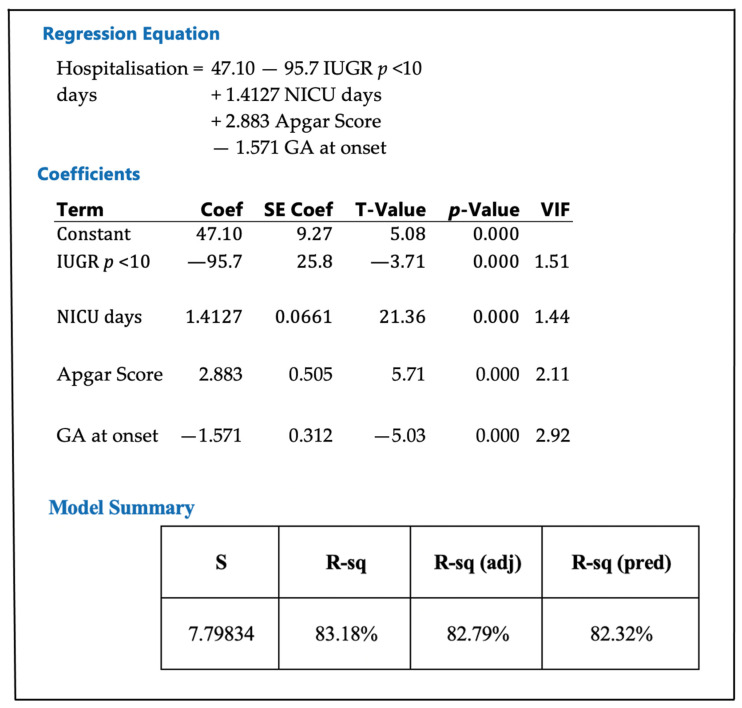
The relationship between hospitalization days and newborn percentile less than percentile 10 (IUGR *p* < 10), neonatal intensive care unit (NICU) days, Apgar score, and gestational age (GA) at delivery; data analyzed using Pearson correlation, *p* < 0.001, R-sq = 83.18%.

**Table 1 medicina-59-00773-t001:** General maternal characteristics and main associations in pregnancies complicated with early- and late-onset IUGR.

Feature/Association	Eo-IUGR	Lo-IUGR	*p*
Age	28.17 (19–37)	29 (17–42)	0.0976
BMI	24.2 (19–28)	23 (18–35)	0.205
Smoking	65.38%	33.9%	<0.01
Gestational hypertension Preeclampsia Eclampsia HELLP Syndrome	75% 44.24% 0 5.78%	25.44% 15.26% 3.4% 1.7%	<0.01 <0.01 <0.01 0.557
GDM	17.31%	5.1%	0.007

Abbreviations: IUGR—intrauterine growth restriction; Eo—early onset; Lo—late onset; BMI—body mass index; HELLP—Hemolysis, Elevated Liver enzymes and Low Platelets; GDM—gestational diabetes mellitus.

**Table 2 medicina-59-00773-t002:** Ultrasound features in pregnancies complicated with early- and late-onset IUGR.

Characteristics	Eo-IUGR	Lo-IUGR	*p*
ST placental UtA centile	98.5 (98–100)	66 (56–100)	<0.01
ST non-placental UtA centile	96.5 (94–100)	51 (28–100)	<0.01
TT placental UtA centile	96 (95–100)	65 (52–100)	<0.01
TT non placental UtA centile	92.3 (91–100)	53 (22–99)	<0.01
TT placental notch	80.5%	9%	<0.01
TT non-placental notch	69.44%	11%	<0.01
TT CPR centile	1 (1–5)	14 (1–39)	<0.01
ST AC centile	9% (1–12%)	35% (10–55%)	<0.01
TT AC centile	1% (1–7%)	20% (1–30%)	<0.01
ST EFW centile	10% (5–15%)	36% (10–56%)	<0.01
TT EFW centile	2% (1–6%)	6% (1–15%)	<0.01

Abbreviations: IUGR—intrauterine growth restriction; Eo—early-onset; Lo—late-onset; ST—second trimester; UtA—uterine artery; TT—third trimester; AC—abdominal circumference; EFW—estimated fetal weight; GA—gestational age; CPR—cerebroplacental ratio. ST data processed were obtained at 22–23 WG for UtA and AC. TT data processed were obtained at 28–29 WG for UtA and notching. In the analysis, the values for CPR centile, AC centile, and EFW were collected last during the pregnancy.

**Table 3 medicina-59-00773-t003:** Comparison between gestational age at delivery and the total number of prenatal visits and visits in the third trimester in the two groups.

Parameter	Eo-IUGR	Lo-IUGR	*p*
GA at delivery	31.7 (28–34)	37 (34–38)	<0.01
Placental weight (g)	312 +/− 109	403 +/− 133	<0.01
Total nr of prenatal visits	11.5 (10–30)	9 (3–18)	<0.01
Nr of prenatal visits in TT	6 (5–15)	4 (2–10)	<0.01

Abbreviations: IUGR—intrauterine growth restriction; Eo—early-onset; Lo—late-onset; GA—gestational age; g—grams; TT—third trimester.

**Table 4 medicina-59-00773-t004:** Newborn data in pregnancies complicated with IUGR.

Characteristic/Complications	Eo-IUGR	Lo-IUGR	*p*
Hospitalization Days	38 (22–90)	3.8 (2–32)	<0.01
Apgar Score	5.3 (1–8)	8.1 (2–10)	<0.01
Resuscitation	19 (36.53%)	7 (11.86%)	<0.01
Birth Percentile	2.3% (1–10%)	7% (3–10%)	<0.01
NICU Days	10.8 (0–60)	2,3 (0–31)	<0.01
Respiratory Distress Syndrome	20 (38.4%)	3 (5.08%)	<0.01
Bronchopulmonary Dysplasia	1 (1.92%)	0	-
Transient Apnea	49 (94.23%)	5 (8.47%)	<0.01
Hypotension	6 (11.53%)	3 (5.08%)	<0.01
PDA	16 (30.76%)	1 (1.69%)	<0.01
IVH	6 (11.53%)	2 (3.38%)	<0.01
PVL	2 (3.84%)	0	-
Hypothermia	0	0	-
Immature GI System	40 (76.92%)	14 (23.72%)	<0.01
NEC	1 (1.92%)	0	-
Anemia	52 (100%)	6 (10.16%)	<0.01
Jaundice	15 (28.84%)	5 (8.47%)	<0.01
Transient Hypoglycemia	22 (42.3%)	3 (5.08%)	<0.01
Infection	12 (23.07%)	1 (1.69%)	<0.01

Abbreviations: IUGR—intrauterine growth restriction; Eo—early-onset; Lo—late-onset; NICU—neonatal intensive care unit; PDA—persistent ductus arteriosus; IVH—intraventricular hemorrhage; PVL—periventricular leukomalacia; GI—gastrointestinal; NEC—necrotizing enterocolitis. The Apgar score and the number of NICU days are expressed as median.

## Data Availability

All data presented here are available from the authors upon reasonable request.

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
