# Peer review of "Intrauterine Growth Restriction—Prediction and Peripartum Data on Hospital Care"

_medicina, 2023, doi:10.3390/medicina59040773_

Round 1

Reviewer 1 Report

The presented study aimed to assess the US detection rates of IUGR and to describe the prenatal care pattern and to establish predictors for the number of total hospitalization days needed postnatally.

The article is well structured according to the authors instructions. 

Please paraphrase the lines between 279 till 283 as per common rules for citing. 

Author Response

The presented study aimed to assess the US detection rates of IUGR and to describe the prenatal care pattern and to establish predictors for the number of total hospitalization days needed postnatally.

The article is well structured according to the authors instructions. 

We respectfully thank reviewer 1 for his/hers comments and suggestions, and for the time invested to review our manuscript.

Please paraphrase the lines between 279 till 283 as per common rules for citing. 

We thank him/her for the opportunity to correct the issue raised.

We corrected the sentences according to the Guidelines for Citations and References (https://www.boisestate.edu/cobe/cobe-writing-style-guide/citations-and-references/).

See lines 289- 293 in the revised form of the manuscript.

Reviewer 2 Report

Review report for autors:

 A brief summary:

The aim of this study is to assess detection rates of IUGR fetuses (the performance of US in estimating the actual neonatal birth weight) to establish predictors in estimating the number of total hospitalization days required postnatally.

The author in this study demonstrated tahat early and late-onset IUGR have different US features, prenatally and postnatal outcomes are different. If the lower percentile of US EFW, the prenatal diagnosis is in that case more likely to be reached and closer follow-up is offered in their department.  The conclusion after the conducted research by the author is that the total number of hospitalization days may be predicted using intrapartum and immediate postnatal data, in both tested groups (early and late IUGR).

With this study, the parameters were studied and a generally logical and accepted conclusion was made, that the more significant and earlier the occurrence of IUGR, the greater the number of hospitalizations and the greater the supervision of such pathological pregnancies.

This study is oriented towards an assessment report mostly, assuming the financial costs, of this clinical diagnosis, you did not specify whether there is a similar study already conducted. Perheps you colud provide more examples of compariation between yours and recent similar research.

General concept comments:

This study listed limitations in terms of unconfirmed US expertise of primary gynecologists, not listed other limitations that I think should have been explored in more detail, as well as potential further researchs.

The title is not entirely well written, it is not concise. I believe that it should be summarized and written more concisely, to point to further text in a targeted manner.

The summary is partially well writen and contains all the important results.

In line 26. does not specify which intrapartum parameters are being considered.

In line 27. it is not clearly stated which department is conducting the survey, the same is stated only in the introduction.

In line 32. citing the analysis of parameters by retrospective regression analysis, it is unclear what was wanted to be achieved.

In line 33. it is stated in the summary that the study was conducted on 111 pregnant women, it is not emphasized in what period of time, the same is explained in the introduction. I think it should be recorded in the summary as well.

In line 34,35 „differents of US“ were recorded in the text, the same was the tool, the criteria of what is wanted to be achieved in the summary are vaguely explained.

The conclusion of the summary is academically meaningless, the above logically implies, the commercial effects of said clinical diagnosis that were the primary goal of the research are not emphasized.

The introduction is clearly and interestingly written, with minor additions it provides an appropriate overview and introduction to the main topic of the paper.

In line 76. it is not entirely clear to the reader whether the text refers to the perinatal or postnatal period. In line 79. of "detecting smallnes in neonates", I do not understand the formulation within the context of the said.

The methods are examplary written and reproducible.

The methodology is clearly and thoroughly written. I'm interested in further clarification as to, why you stated in line 102. that it was tracked as US paramether PI, not RI in the early IUGR tracking? Does your research use DV as a parameter? By what guidelines did you exclude DV? The same advises ISUOG, which you referred to in other parameters (you also specified that in row 289.)

The results are adequately presented. I think the results require only small changes. In all tables, the level of significance should be indicated and the conduct of statistical analysis should be recorded. In line 185. reduce the spacing. The results are clear, systematic, easy to interpret, as well as schematic representations. In line 255.,256. should reduce the font of letters, currently the size is the same as the meaning of the table.

The discussion is partially clear and concise. The statemensts in the discussion are drown coherently and are supported by appropriate citations. The discussion itself is specifically written, but in some parts such as line 359. it is not clear whether the statement conclusion of the researcher or the reference. I also think, it needs to be better explained. The same applies to MCA or CPR , it was concluded that they differ significantly, but not in what sense. Which parameter is more significant? The sentence is sketchy, I think it needs clarification.

The literature is correctly cited. Slightly more than 43% of references (16/37) are references of recent publications. I believe that references should be included in the paper, as well as data from similar financial research conducted on similar clinical diagnoses. And that it is necessary to insert and refer to the general conclusions of financial management in the vulnerable group of neonates.

The conclusion is consistent ,consize with clearly presented arguments.

The hypothesis of the article is clearly written (the aim of the study), explained and later developed methodologically. Experimental design of the study is appropriate for the testing inicial hypothesis of the study.

This study is a new way of seeing and monitoring pathological pregnancies in the time period of financial crises and inflation that surround us. Applicable to a smaller number of specialists, especially those employed in tertiary center administrations.

In conclusion, I belive that this is an interesting work ,with a new perspective in the views of clinicians applicable in a narrower, strictly defined population (menagement of clinical hospitals), and I recommend its acceptance for publications after minor corrections.

Author Response

We respectfully thank reviewer 2 for his/hers comments and suggestions. We highly appreciate the time to review our manuscript, and the amount of work invested to improve the manuscript.

We thank him/her for the opportunity to clarify/correct the issues raised.

  • This study is oriented towards an assessment report mostly, assuming the financial costs, of this clinical diagnosis, you did not specify whether there is a similar study already conducted. Perhaps you could provide more examples of comparison between yours and recent similar research.

We thank the reviewer for raising this issue.  We performed repeated searches in PubMed, Google Scholar, Scopus, Web of Science, Science Direct, PsycINFO, and ERIC databases for English language studies, and we could not find any similar study regarding the financial costs or hospitalization days following the diagnosis of IUGR.

  • This study listed limitations in terms of unconfirmed US expertise of primary gynecologists, not listed other limitations that I think should have been explored in more detail, as well as potential further research.

We thank the reviewer.

We apologize, we were not able to avouch any other limitations. The COVID pandemic has disrupted our medical care system, by decreasing patient addressability to outpatient care.

Thus, we admit that the total figures are still low. As potential further research, we intend to conduct a larger prospective study, including a significantly larger number of patients, studied for a longer period. This will include long-term follow-up of the children enrolled in the study as fetuses.

We added this statement in the revised manuscript, in the discussion section.

  • The title is not entirely well written, it is not concise. I believe that it should be summarized and written more concisely, to point to further text in a targeted manner.

We thank the reviewer for the suggestion. We edited the title accordingly. We hope it reflects the manuscript content appropriately.

  • The summary is partially well-written and contains all the important results. In line 26. does not specify which intrapartum parameters are being considered.

We thank the reviewer for the suggestion. Unfortunately, we had to comply with MDPI Instructions for Authors, by which the abstract should be 200 words maximum.  Adding any other information would further exceed this limit (we were not able to shorten the current revised form of the abstract below 265 words).

  • In line 27. it is not clearly stated which department is conducting the survey, the same is stated only in the introduction.

We added the information in the abstract.

  • In line 32. citing the analysis of parameters by retrospective regression analysis, it is unclear what was wanted to be achieved.

We edited the last phrase accordingly, to give a more comprehensive conclusion in the abstract.

  • In line 33. it is stated in the summary that the study was conducted on 111 pregnant women, it is not emphasized in what period of time, the same is explained in the introduction. I think it should be recorded in the summary as well.

We added the information in the abstract.

  • In lines 34,35 „different of US“ were recorded in the text, the same was the tool, and the criteria of what is wanted to be achieved in the summary are vaguely explained.

We appreciate the suggestion. As explained above, the limit of maximum words corresponding to the abstract had to be met. We detailed this information in the material and methods section.  

  • The conclusion of the summary is academically meaningless, the above logically implies the commercial effects of said clinical diagnosis which were the primary goal of the research are not emphasized.

We edited the last phrase accordingly.

  • The introduction is clearly and interestingly written, with minor additions it provides an appropriate overview and introduction to the main topic of the paper. In line 76. it is not entirely clear to the reader whether the text refers to the perinatal or postnatal period.

Thank you for your suggestions. We edited the phrase.  

  • In line 79. of "detecting smallness in neonates", I do not understand the formulation within the context of the said.

We used the Hadlock formula for the estimated fetal weight (EFW) and noted the corresponding percentile. For interpreting the neonatal weight percentile, we used reference data from the INTERGROWTH-21(st) Project and the calculators offered by https://www.omnicalculator.com/health/birthweight-percentile website.

We used these data to compare the EFW percentile with the birth percentile.

  • The methods are exemplary written and reproducible. The methodology is clearly and thoroughly written. I'm interested in further clarification as to, why you stated in line 102. that it was tracked as US parameter PI, not RI in the early IUGR tracking?

At the beginning of our study, we based the development the study protocol on the most recent guidelines published back then. https://obgyn.onlinelibrary.wiley.com/doi/10.1002/uog.20272

(Salomon LJ, Alfirevic Z, Da Silva Costa F, Deter RL, Figueras F, Ghi T, Glanc P, Khalil A, Lee W, Napolitano R, Papageorghiou A, Sotiriadis A, Stirnemann J, Toi A, Yeo G. ISUOG Practice Guidelines: ultrasound assessment of fetal biometry and growth. Ultrasound Obstet Gynecol. 2019). The consensus-based definitions for early and late fetal growth restriction in absence of congenital anomalies include UA-PI centile and UtA-PI centile.

  • Does your research use DV as a parameter? By what guidelines did you exclude DV? The same advises ISUOG, which you referred to in other parameters (you also specified that in row 289.)

We did not include the DV in our parameters research. DV Doppler alteration (absent or reversed flow) is a final predictive parameter, which requires emergency delivery. The citation used is subsequent to our study's beginning date.

  • The results are adequately presented. I think the results require only small changes. In all tables, the level of significance should be indicated, and the conduct of statistical analysis should be recorded.

The statistical analysis is described in lines 157-162. We added the level of significance.

  • In line 185. reduce the spacing.

Thank you. We reduced the space.

  • The results are clear, systematic, easy to interpret, as well as schematic representations. In lines 255.,256. should reduce the font of letters, currently the size is the same as the meaning of the table.

We reduced the font of the letters.

  • The discussion is partially clear and concise. The statements in the discussion are drawn coherently and are supported by appropriate citations. The discussion itself is specifically written but in some parts such as line 359. it is not clear whether the stated conclusion of the researcher or the reference. I also think it needs to be better explained. The same applies to MCA or CPR , it was concluded that they differ significantly, but not in what sense. Which parameter is more significant? The sentence is sketchy, I think it needs clarification.

We rephrased the paragraph.

  • The literature is correctly cited. Slightly more than 43% of references (16/37) are references of recent publications. I believe that references should be included in the paper, as well as data from similar financial research conducted on similar clinical diagnoses. And that it is necessary to insert and refer to the general conclusions of financial management in the vulnerable group of neonates.

We thank you for the suggestion. As mentioned above, we were not able to find any similar study regarding the financial costs or the number of hospitalization days following the diagnosis of IUGR.

Currently, we are processing in our hospital the financial data arising from our results. The consequences confirmed so far are reported. We found that using the birth weight, the number of NICU days, the Apgar score, and the GA we can predict 83% of the variance of hospitalization days needed by these neonates postnatally. This finding helped us to estimate better the number of neonatal units becoming available at a certain time point. We are not able to draw any other definite conclusions so far.